# Kuhn–Munkres Algorithm-Based Matching Method and Automatic Device for Tiny Magnetic Steel Pair

**DOI:** 10.3390/mi12030316

**Published:** 2021-03-18

**Authors:** Zheng Xu, Guo-zhao Yuan, Xiao-dong Wang, Xian-shuai Quan, Tong-qun Ren, Jun-shan Liu

**Affiliations:** Key Laboratory for Micro/Nano Technology and System of Liaoning Province, Dalian University of Technology, Dalian 116024, China; yuanguozhao@mail.dlut.edu.cn (G.-z.Y.); xdwang@dlut.edu.cn (X.-d.W.); Mr_quan_xs@mail.dlut.edu.cn (X.-s.Q.); ren_tq@dlut.edu.cn (T.-q.R.); liujs@dlut.edu.cn (J.-s.L.)

**Keywords:** tiny magnetic steel pair, Kuhn–Munkres algorithm, matching method, microassembly

## Abstract

The tiny magnetic steel pair (TMSP), composed by two tiny magnetic steel blocks (TMSBs), is critical for some precision instruments. Incorrect matching of TMSP may result in insufficient instrument performance. Herein, the matching method of TMSP based on the Kuhn–Munkres algorithm is proposed. Further, an automatic TMSP matching device is developed. Especially, an ingenious clamp for multiple constraints of TMSB is presented and a visual/magnetism/force hybrid control strategy is realized for the safe and efficient manipulation of TMSBs in a magnetic environment. Moreover, with the TMSBs of a pendulum accelerometer, the matching experiments are conducted to validate the comprehensive performance. The result of the numerical experiment shows that the Kuhn–Munkres algorithm-based method is stable and efficient. The results of measurement and TMSP matching experiments show that the device has good repeatability (<1 mT) and practicability. The proposed matching method has great application prospect in various matching and microassembly of TMSPs.

## 1. Introduction

The tiny magnetic steel pair (TMSP), composed by one tiny magnetic steel block (TMSB) and another TMSB as an opposite pole, is critical for some precision instruments such as the accelerometer, three-floated gyroscope, small motor etc. [1,2,3]. The TMSP function is to generate an even magnetic field and correspondingly apply ampere force on coil or other objects in an air-gap, inducing them to deflect, translate, keep balance with external forces etc. [4]. Obviously, the magnetic field is dependent on not only the size and position of TMSBs, but also the magnetizing parameters. Therefore, to obtain satisfied TMSP for the followed assembly, the selective TMSP matching from a certain amount of TMSBs based on magnetic flux density is indispensable [5,6]. In related industrial areas, most TMSP matching processes are manually operated with precise mechanical fixtures. The subtle wear of fixtures and the unavoidable finger trembling might seriously affect the consistency. Moreover, unexpected demagnetization often happens, owing to the personal error. Therefore, it is urgent to develop an automatic method for TMSP matching.

The generalized process of TMSP matching can be divided into three steps:Data collection. The size and surface magnetic flux density of TMSBs (n-TMSB and s-TMSB mean the TMSBs as the N-pole and S-pole of an air-gap magnetic field) are measured and saved.Matching process. According to the surface magnetic flux density of n-TMSBs and s-TMSBs, those qualified TMSBs are preliminarily screened out and estimated to make sure of the possibility of TMSPs. As a result, the likely suitable TMSPs will be found out (here ‘suitable’ means that their air-gap magnetic flux density is high enough).nResult verification. Based on the estimated combinations, the qualified TMSBs are assembled into the corresponding magnet bases to form TMSPs. Then, the air-gap magnetic field in each TMSP is measured to verify the performance.

Although there is no direct report about TMSP matching, some related works exist. For example, Meyer et al. [7] proposed an automated logistics and storage solution for TMSBs of permanent magnet motor. This solution improved the traceability of single TMSBs and enabled to compensate variations of the magnetic properties by selective TMSB assembly. Arbenz et al. [8] presented an approach to deduce the magnetization of TMSB. By using the Hall sensor to scan the magnetic field, the TMSB magnetization could be inferred. Franke [9] presented an assembly solution for magnetized high coercive TSMP, including an automated TMSB separator and some different grippers. More recently, the patent proposed by Apple Inc. [10] presented a device for automatic measurement of TMSBs, in which a precision turntable was used to achieve intermittent feed movement of TMSBs. Moreover, several Helmholtz coils were used to measure the magnetic flux density of TMSB and to determine whether the property of TMSB was qualified. But until now, as far as we know, the complete research on the automatic matching method had not yet been reported. The main challenges of TMSP matching lie in: Firstly, the suitable method needs to be developed to predict the magnetic flux density of TMSPs and to optimize its combination [11,12]. Secondly, the strong magnetic forces might cause demagnetization or damage of TMSBs due to collision in matching process [13]. Thirdly, the air-gap magnetic flux density distribution is highly sensitive to assembly accuracy [14,15]. Obviously, if the TMSB is incorrectly assembled, it will affect the matching performance.

Here, inspired from the Kuhn–Munkres (K-M) algorithm, a novel TMSP matching method for TMSP is proposed. Additionally, the numerical experiment is performed to verify the reliability and efficiency of the method. Further, an automatic TMSP matching device using the matching method is developed too. Especially, an ingenious clamp for multiple constraints of TMSB and a visual/magnetism/force hybrid control strategy are realized for the safe and efficient manipulation of TMSBs in a complex magnetic environment. Finally, utilizing the TMSBs from a type of pendulum accelerometer, the matching experiments are actually conducted to validate the comprehensive performance.

## 2. TMSP Matching Method

### 2.1. Principle of Matching Algorithm

The TMSP matching can be considered the weighted bipartite graph matching problem. Herein, the K-M algorithm about the optimal matching of the bipartite graph is utilized. In principle, by giving each vertex a top mark, the algorithm can convert the problem of maximizing weight matching into continuous searching an augmentation path to make the bipartite graph [16,17]. By inputting the predicted value of *B_a_* for each combination as the edge weight, the optimal combination can be obtained. The matching process will not stop until the number of matched pairs reaches the number of maximum complete matching. The matching principle for TMSP can be described as shown in Figure 1.

Magnetostatics modeling. Through FEM simulation, the distribution of magnetic flux density in TMSP is calculated. Then, an equation is built up to describe the influence of magnetic flux densities of s-TMSBs and n-TMSBs (*B_S_* and *B_N_*) on the air-gap magnetic flux density (*B_a_*).Weight calculation. *B_S_* and *B_N_* are measured firstly. Then, the n-TMSBs and s-TMSBs are numbered as (*S_i_*, *N_j_*) respectively for the left vertex and right one of bipartite graph. $B_{a}\left(i,\;j\right)$ is calculated by the *B_Si_* and *B_Nj_*, as the weight of edge connected by each vertex.Vertex assignment. It means that the maximum edge weight (max [$B_{a}\left(j\right)$]) of all edges connected to each *S* vertex on the left is assigned to the top mark (*L_S_*). Additionally, the top mark (*L_N_*) of each N vertex on the right is assigned to zero.TMSP matching. The algorithm starts from the vertex *S*_1_ on the left to search for the corresponding vertex and augmented path (*N* vertex). The guideline of matching is to keep only the edges with the same weight $B_{a}\left(i,\;j\right)$ and the left top mark *L_Si_*, and to meet the requirements of $L_{Si}{\;+\;L}_{Nj}\;\geq {\;B}_{a}\left(i,\;j\right)$. If one edge is not qualified or the two edges conflict, then *L_S_* of all left vertices of the conflict path is subtracted by a top mark adjustment *d* (*d =* min [$B_{a}\left(i,\;j\right)$ − ($\;L_{Si}{\;+\;L}_{Nj}$)]), and *L_N_* of all right vertices is increased by *d*. After that, the pairing is performed again, and the augmented path is searched until the maximum matching number (min [*i*, *j*]) is reached.Result verification. According to the matching result, s-TMSB is actually assembled with n-TMSB to form a TMSP and then *B_a_* is measured to verify it.

The complexity of the K-M algorithm for the TMSP matching problem is as follows:

Suppose $n=\max\left[i,\;j\right]$, then the number of edges *m* (the number of all TMSP combinations) is at most *n^2^*. When searching for an augmented path in step 4, both deep first search (DFS) and breath first search (BFS) can be used. The K-M algorithm is based on the BFS method. By increasing the intermediate array to record the change amount *d* of the top mark, the repeated search of the augmented path can be reduced. The complexity of a BFS operation is *O* (*n*), so the maximum complexity of the K-M algorithm applied is O (*n* × *n*^2^ = *n*^3^) [18,19].

### 2.2. FEM Simulation

The data required for magnetostatics modeling is obtained by FEM. We choose the TMSP framework from a pendulum accelerometer as shown in Figure 2. When the acceleration (*a*) is input into the pendulum accelerometer, relative displacements (deflection angle *θ*) of the inertial pendulum will appear. Then, the current *I* being proportional to *θ* is transmitted to the torque coil connected to the pendulum by the sensor and corresponding circuit. In the air-gap magnetic field (*B*), a balance moment *M_f_* is generated to balance the inertial pendulum energized coil. It means that the air-gap magnetic field (*B*) can directly affect the output of the pendulum accelerometer [20].

The TMSP is mainly composed of a s-TMSB and a base with a glued n-TMSB (N-pole of the n-TMSB and S-pole of the s-TMSB are set oppositely, and magnetized direction is *Y*-direction). With COMSOL software (COMSOL Inc. Stockholm, Sweden), the simulation of magnetic field distribution is carried out. Related parameters are shown in Table 1.

The magnetic flux density distribution is shown in Figure 3a. The influence of *B_r-S_* and *B_r-N_* on $B_{a}$ (point in the middle) is shown in Figure 3b. $B_{a}$ is basically increased with the increase of the remanence of the two TMSBs (*B_r-S_* and *B_r-N_*) that can be described by Equation (1). For this TMSP of accelerometer, *a* is 0.502 and *b* is 0.192:(1)$$B_{a}=aB_{r-S}+bB_{r-N}.$$

Since here the TMSB is a rectangular permanent magnet, the relationship between the remanence *B_r_* and the magnetic flux density *B* at certain distance *x* from the magnet surface in a magnetized direction can be described [21,22]:(2)$$B_{r}=\frac{\pi }{{tan}^{-1}(\frac{LH}{2x\sqrt{L^{2}+H^{2}+4x^{2}}})-{tan}^{-1}(\frac{LH}{2\left(x+W\right)\sqrt{L^{2}+H^{2}+4{\left(x+W\right)}^{2}}})}B=\gamma B,$$
where *L*, *H*, *W* are the length, height, and width of magnet, in which *W* is in the magnetized direction. For this same kind of TMSB, α is fixed to 2.939. Thus, a′ = γ ⋅ a = 1.688, b′ = γ ⋅ b = 0.645 in Equation (3). The equation has been verified by Gauss meter and the deviation is less than 1.8%:(3)$$B_{a}=a\prime B_{S}+b\prime B_{N}.$$

## 3. TMSP Matching Device

### 3.1. Device Structure

An automatic device for TMSP matching is developed as shown in Figure 4, which is composed of the TMSB fixture module, the single-TMSB measurement module, the TMSP matching module, and the microscopic imaging module, being installed on a 3D linear precision motorized stage (Repeatability: ±1 μm). These parts that directly contact with TMSBs and magnet bases are made of brass to avoid magnetization. Owing to modular design, the device can be easily modified for various TMSBs.

The TMSB fixture module is used to fix the magnet bases and s-TMSBs. At present, it is composed of five fixtures. Every s-TMSB is placed in one socket in which the space on its left and right is kept for the TMSB clamp fingers.The single-TMSB measurement module consists of two Hall probes (Resolution: 10 nT, range: 30 T), air slide table, and corresponding connecting parts, which are used to measure the magnetic flux density of the n-TMSBs and the s-TMSBs, respectively.The microscopic imaging module (1×, CCD resolution: 3840 × 2748, Pixel size: 1.67 μm) is used to measure the size and position of TMSBs and magnet bases.The TMSP matching module consists of the TMSB clamp, air gripper, Hall probe, flexible mechanism, micro-force sensor, and connecting parts as shown in Figure 5.

The TMSB clamp is used to clamp TMSB. It is composed of two fingers driven by an air gripper. The structure of the finger hook is used to prevent the clamped TMSB from falling off by magnetic force induced by other TMSBs. In the clamping state, TMSB is clamped by the pressure of the side clamping surface on the side surface of the clamp. The lower surface of the finger hook structure contacts the lower surface of TMSB.

A micro-force sensor is used to detect the force in the matching process to judge the contact state between the clamp and TMSB. It is connected to the air gripper through a U-shaped leaf spring. According to the experimental result, the threshold is set to 0.2 N. With the information of contact state, the excessive squeeze in *Z*-direction on the clamp and TMSB can be avoided. The front end of the clamp is equipped with a Hall probe, which can measure in situ the air-gap magnetic flux density ($B_{a}$) in TMSP.

Complete process of TMSP matching with the device is as follows:TMSB size and position measurement. The size and position are measured via the microscopic imaging module. These parts with qualified size will be put into a list.TMSB magnetic property measurement and matching prediction. The single-TMSB measurement module is driven to measure the magnetic flux density *B_S_* and *B_N_* in the list respectively. Then, the above-mentioned matching method is used to predict and give the list of matching result.TMSP matching. The TMSP matching module is driven to the top of first s-TMSB in the matching list, and then the clamp is controlled to the position and pick up the s-TMSB. Then, the clamped s-TMSB is moved to the matching position and temporarily fixed on the magnet base. Additionally, $B_{a}$ of TMSP is measured by the Hall probe in front of the TMSB clamp.

With the magnetic flux density $B_{a}$ within the required qualified range, the clamp will release the s-TMSB and keep it on the magnet base. Otherwise, the s-TMSB will be moved to the next matching position of magnet base. If the s-TMSB cannot be matched after whole traversal, it will be put back to the initial position.

### 3.2. Control Strategy for TMSB Positioning Task

To make the safe and efficient positioning of TMSB in a complex magnetic environment, a visual/magnetism/force hybrid control strategy is proposed as shown in Figure 6. After the TMSP matching module is driven to the top of the first s-TMSB in list, the clamp is controlled to the position and then picks up the s-TMSB. The spatial positions of s-TMSBs and magnet bases are obtained via the microscopic imaging module. According to those photos, the coarse positioning of s-TMSB and the magnet base is realized. Using force servo and magnetic servo, the fine positioning is completed by controlling the feedback value of contact force and magnetic flux density.

The micro-force sensor is integrated to obtain the contact force during the lowering of the clamp to determine whether it has reached the expected position. Once the contact force exceeds a given threshold (0.2 N), the 3D linear motorized stage stops the *Z*-direction displacement to ensure the safety of the Hall probe and the TMSBs. Then, the clamp is driven close to the TMSB along the negative *Y*-direction.

Once the magnetic flux density change measured by the Gauss meter is less than a threshold (1 mT) during the process, the 3D linear stage stops and the TMSB is fully positioned. After that, the air gripper is closed to clamp the TMSB. Then, the clamped s-TMSB is moved to the matching position of the magnet base.

## 4. Experiments

The TMSB measurement experiment, numerical experiment, and TMSP matching experiment are carried out. Before the experiments, the Gauss meter and the Hall probes are checked by professional verification organization (measuring range: 0.04–2.00 T, uncertainty: $U_{rel}{\;=\;2\;\times \;10}^{–5},\;k\;=\;2)$.

The magnetic flux densities of TMSBs are measured with the visual/magnetism/force hybrid control strategy. Each TMSB is measured five times. The results of two s-TMSBs and two n-TMSBs are shown in Table 2. The average time for each TMSB measurement is ~15 s. The result shows that the device presents good repeatability (<1 mT) and the control strategy can guarantee the consistency.

To further verify reliability and efficiency of the matching method, the numerical experiment is carried out with MATLAB. Different combinations of *B_S_* and *B_N_* are carefully selected (*B_S_* range: 220~275 mT, *B_N_* range: 130~170 mT), and data are combined to iterate 1500 pairs. Generally, the higher average $\bar{B_{a}}$ of all matched TMSP in qualified range (490~550 mT) means that the matched combinations are more appropriate. As shown in Figure 7, the average value of air-gap magnetic flux density ($\bar{B_{a}}$) by the K-M algorithm-based method can be basically stabilized at about 515 mT. When the number of matched pairs is small (<500), the running time is within 5 s. These results show that the method presents an advantage in multi-group matching, and is efficient and stable for the matching of an amount of TMSBs in production.

By using the developed device, five sets of s-TMSBs and magnet bases are installed to match for the experiment. The actual matching results are shown in Table 3. The qualified range of *B_S_* here is 240~270 mT. A negative *B_S_* means that the TMSB is installed in the wrong direction, and a zero value means that no TMSB is installed in this position. The qualified range of *B_N_* is 140~155 mT, and the qualified range of *B_a_* is 490~550 mT. As shown in Table 3, $B_{a}$ matched by this device is within the required qualified range, and $\bar{B_{a}}\;$is equal to 525 mT. These results show that the device can accurately match more suitable TMSPs from multiple sets of TMSBs and can successfully complete the microassembly of TMSB. The matching method and device can be modified for various TMSBs, which has great prospect in various related-magnetism microassembly tasks.

## 5. Conclusions

In this paper, a Kuhn–Munkres algorithm-based method and an automatic TMSP matching device are proposed to selectively match TMSPs from TMSBs. Especially, an ingenious clamp for multiple constraints of TMSB and a visual/magnetism/force hybrid control strategy are realized for safe and efficient manipulation of TMSBs in a magnetic environment. The result of the numerical experiment shows the reliability and efficiency of the Kuhn–Munkres algorithm-based matching method. The results of the measurement experiment and TMSP matching experiment show that the device has good repeatability (<1 mT) and practicability. The proposed matching method and automated device has great prospect in various magnetism-related microassembly tasks.

## Figures and Tables

**Figure 1 micromachines-12-00316-f001:**
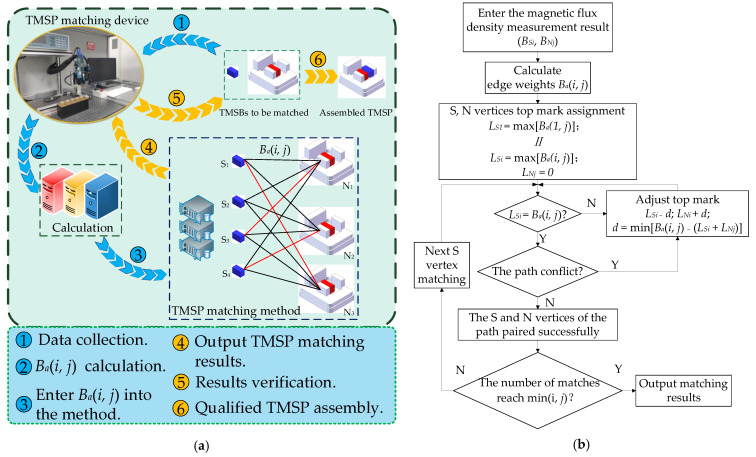
Tiny magnetic steel pair (TMSP) matching method. (**a**) Diagram; (**b**) Flow chart.

**Figure 2 micromachines-12-00316-f002:**
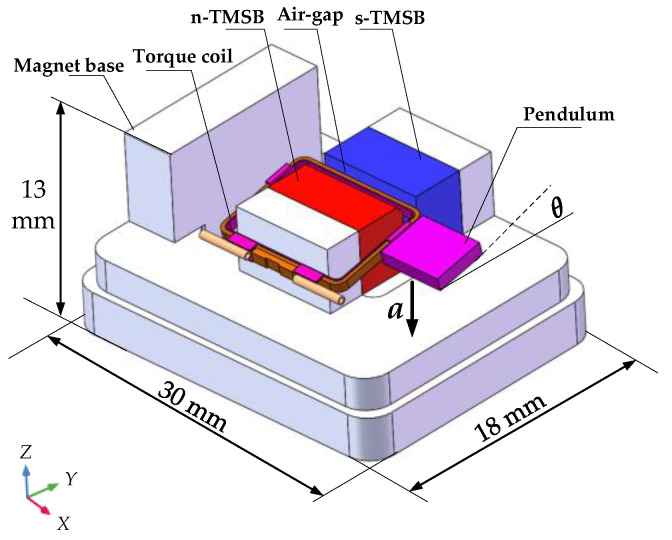
Schematic drawing of pendulum accelerometer.

**Figure 3 micromachines-12-00316-f003:**
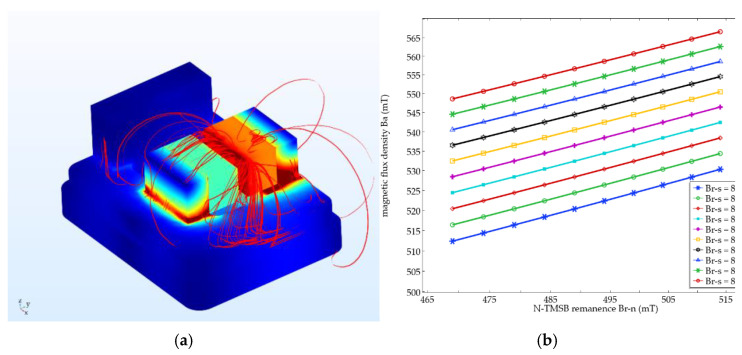
Simulation results. (**a**) Magnetic flux density distribution; (**b**) Influence of *B_r-S_* and *B_r-N_* on $B_{a}$

**Figure 4 micromachines-12-00316-f004:**
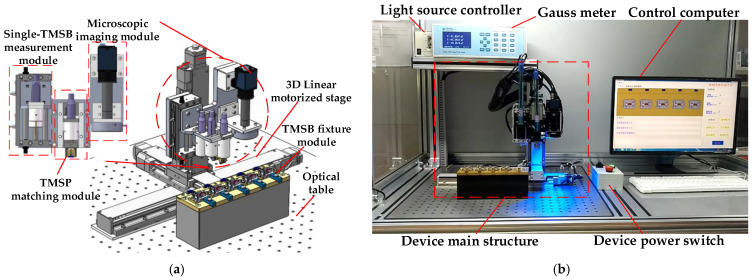
TMSP matching device. (**a**) Overall structure; (**b**) Device in matching process.

**Figure 5 micromachines-12-00316-f005:**
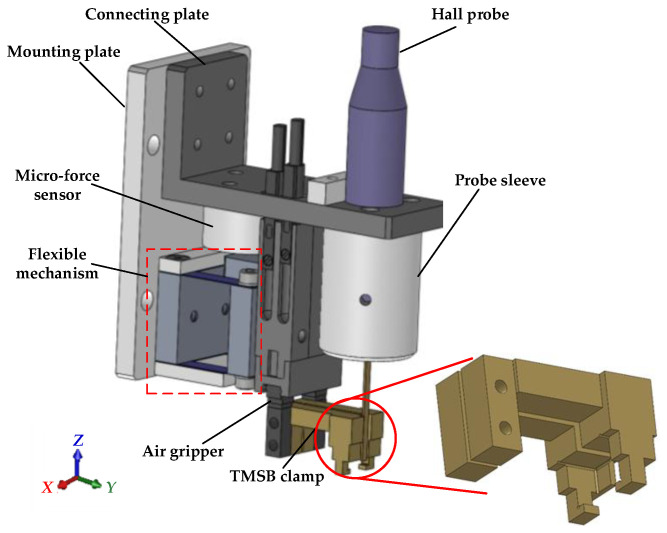
Schematic diagram of the TMSP matching module.

**Figure 6 micromachines-12-00316-f006:**
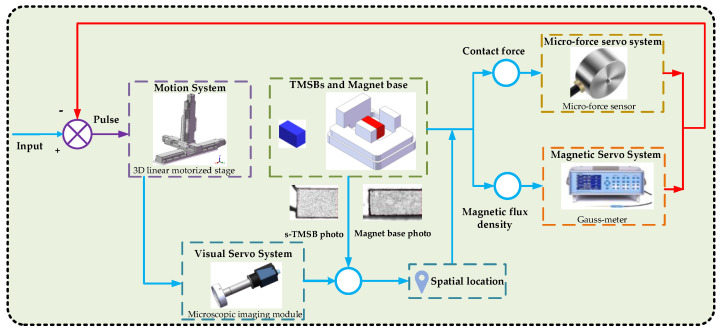
Control strategy for tiny magnetic steel block (TMSB) positioning task.

**Figure 7 micromachines-12-00316-f007:**
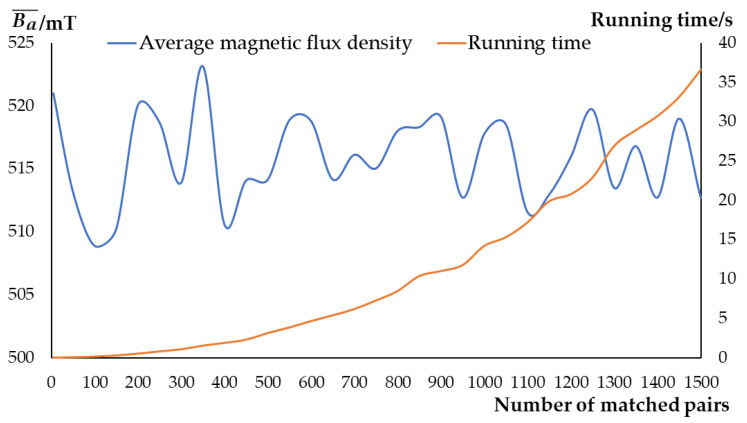
Numerical experiment result.

**Table 1 micromachines-12-00316-t001:** Parameters for simulation.

Object	Material	Relative Permeability	Remanence (mT)	*L × W × H* (mm)
s-TMSB	SmCo alloy	1.03	806~896	8 × 4 × 4.8
n-TMSB	NdFeB alloy	1.05	469~514	8 × 4 × 4.8 (Chamfer)
Magnet base	Soft magnetic alloy	4000	0	-
Air	Air	1	0	(40, 40, 40)

**Table 2 micromachines-12-00316-t002:** Measurement results.

Number	*B_S_*_-1_ (mT)	*B_S_*_-2_ (mT)	*B_N_*_-1_ (mT)	*B_N_*_-2_ (mT)
1	265.92	261.81	152.28	145.58
2	265.92	261.81	152.28	145.58
3	265.37	260.26	152.80	145.37
4	265.92	261.81	152.28	145.58
5	265.37	260.26	152.80	145.37
Standard deviation	0.2694	0.7593	0.2547	0.1029

**Table 3 micromachines-12-00316-t003:** TMSP matching results.

Group	Number	*B_S_* (mT)	*B_N_* (mT)	Method Prediction $B_{a}$ Result (mT)	Actual Measurement $B_{a}$ Result (mT)	$\bar{B_{a}}$ (mT)
1	1	265.37	162.13	(*S*_1_, *N*_5_, 546.67) (S_2_, *N*_3_, 525.55) (*S*_3_, *N*_2_, 537.83) (*S*_4_, *N*_1_, 525.70)	(*S*_1_, *N*_5_, 537.67) (*S*_2_, *N*_3_, 515.56) (*S*_3_, *N*_2_, 536.83) (*S*_4_, *N*_1_, 524.66)	528.69
	2	251.02	152.8			
	3	260.26	157.94			
	4	249.48	145.37			
	5	−213.78	153.14			
2	1	264.59	142.45	(*S*_1_, *N*_5_, 502.67)	(*S*_1_, *N*_5_, 520.91)	527.55
	2	252.25	146.37	(*S*_2_, *N*_4_, 516.18)	(*S*_2_, *N*_4_, 524.84)	
	3	266.17	153.97	(*S*_3_, *N*_3_, 525.34)	(*S*_3_, *N*_3_, 528.47)	
	4	251.21	148.39	(*S*_4_, *N*_2_, 512.73)	(*S*_4_, *N*_2_, 538.69)	
	5	262.68	152.64	(*S*_5_, *N*_1_, 506.75)	(*S*_5_, *N*_1_, 524.84)	
3	1	231.09	156.34	(*S*_1_, *N*_4_, 492.97)	(*S*_1_, *N*_4_, 514.76)	524.8
	2	234.37	145.95	(*S*_2_, *N*_5_, 492.82)	(*S*_2_, *N*_5_, 504.09)	
	3	256.90	150.65	(*S*_3_, *N*_3_, 530.77)	(*S*_3_, *N*_3_, 542.70)	
	4	245.61	159.55	(*S*_4_, *N*_2_, 508.69)	(S_4_, *N*_2_, 526.49)	
	5	253.95	150.76	(*S*_5_, *N*_1_, 529.46)	(*S*_5_, *N*_1_, 531.96)	

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
