# Peer review of "Kuhn–Munkres Algorithm-Based Matching Method and Automatic Device for Tiny Magnetic Steel Pair"

_micromachines, 2021, doi:10.3390/mi12030316_

Round 1

Reviewer 1 Report

The exact purpose of the article is not clear.
It is necessary to specify the goal.
The mathematical basis used is not sufficient.
We know nothing about the Hungarian method
(which is actually the Kuhn-Munkres method).
In the end, the 4.12% difference between the recommended
(but not sufficiently detailed) method and the Hungarian method,
which is also not described.
It is not appropriate to mention the Physical Review Letter article
as a practical source.
The other cited papers only remotely help
to wrap up the basic problem.

Reviewer 2 Report

Please find the review comments in the attachment.

Author Response

Please see the attached file, thanks.

Reviewer 3 Report

This paper proposes a method to automatically select two magnetic blocks with similar magnetic moments to form a pair using a matching method based on the Kuhn-Munkres algorithm. The method is crucial for assembling magnetic blocks for many precision instruments. The authors demonstrated the efficiency of the proposed method with controlled experiments, showing performance improvements in terms of accuracy, repeatability and computational time. The authors also implemented a system to automatically collect and process the data as well as manipulate the magnetic blocks. The manuscript is generally clearly written and well organized. Here are some comments to help further improve the performance.

  1. The overview of the manuscript is that the authors have engineered a system with comprehensive results of data collection, computational matching, and validation. However, the core research contribution needs to be better emphasized to clearly show the challenges that they have resolved using their developed method and system.
  2. The original Kuhn-Munkres algorithm should be first introduced by citing literature work as the authors did not proposed the algorithm but rather applied it for a specific engineering problem. The authors are also suggested to discuss in more details why the Kuhn-Munkres algorithm could yield a better performance in terms of matching accuracy and computational efficiency.
  3. The importance of the improved accuracy. The authors showed that the Kuhn-Munkres algorithm gives about 4.12% higher matching accuracy compared with the classic-Hungarian method. Can the authors discuss why this improvement is significant?
  4. In the experiment, will measure the magnetic field at multiple locations help improve the matching results? In addition, how is the accuracy of the fine positioning of the magnet block relative to the Gauss meter probe?
  5. Please consider introducing the “pendulum accelerometer” briefly to help understand the requirement and importance of assembling magnetic pairs precisely.
  6. In the assembly experiment, how is the magnetic interaction force between magnet blocks handled to ensure positioning accuracy?
  7. The fonts need to be larger in Figure 1, Figure 3(b) and Figure 4(a) to show the annotations clearly. In addition, Figure 4(b) should be annotated clearly.
  8. A video will help illustrate the whole assembly process if possible.

Author Response

请参见附件。

Round 2

Reviewer 1 Report

The authors have made visible efforts to address the issues raised. Defining the topic of the article is now acceptable. Unfortunately, however, I still do not consider the description
of the two algorithms and the detection of their differences to be
well-founded and satisfactorily implemented.
I still believe that a 4.12% error between two similar algorithms
cannot be measured at the algorithmic level in engineering practice,
since e.g. a numerical representation or an uncertainty propagation
can result in higher errors.

Reviewer 2 Report

I'm pleased with the modification. All my concerns have been addressed. 

Author Response

Thanks for your valuable and constructive comments to greatly improve  our manuscript.

Reviewer 3 Report

The authors have addressed my concerns in their revised manuscript. I don't have further comments.

Author Response

(The authors gave the same response as above.)

Round 3

Reviewer 1 Report

My problem is that we didn't understand each other.
I live in Hungary, I am a mathematician and I have been teaching
the Hungarian method at university for more than twenty years.
  The difference in algorithms cannot be derived solely from complexity.
Please show the difference in a table or figure.
Therefore, I ask again to explain now the stability
(previously 4.12% error) by concentrating only on the two algorithms.
I do not consider this stability issue (error) to be significant and cannot stem from
differences in algorithms at all. The basic question is why it is necessary to describe two algorithms at all?
Why it is not enough to just use only one algorithm?
The experimental part of the article is quite strong,
math is always a dangerous plant.

Round 4

Reviewer 1 Report

I can accept this fourth version